# The effects of viewing a winter forest landscape with the ground and trees covered in snow on the psychological relaxation of young Finnish adults: A pilot study

**Ernest Bielinis**[1]*, **Emilia Janeczko**[2], **Norimasa Takayama**[3], **Anna Zawadzka**[1], **Alicja Słupska**[1], **Sławomir Piętka**[1], **Maija Lipponen**[4], **Lidia Bielinis**[5]

**1** Department of Forestry and Forest Ecology, Faculty of Environmental Management and Agriculture, University of Warmia and Mazury in Olsztyn, Olsztyn, Poland, **2** Department of Forest Utilization, Institute of Forest Sciences, University of Life Sciences in Warsaw, Warsaw, Poland, **3** Environmental Planning Laboratory, Department of Forest Management, Forestry and Forest Products Research Institute in Japan, Tsukuba, Japan, **4** Natural Resources Institute Finland (LUKE), Paltamo, Finland, **5** Department of General Pedagogy, Faculty of Social Science, Olsztyn, Poland

* ernest.bielinis@uwm.edu.pl

**Data Availability Statement:** All data files (Bielinis et al. 2020 Data.xlsx) are available from the Mendeley database: Ernest Bielinis(2020),

## Abstract

Forest bathing is an outdoor activity, and it might be a promising preventive treatment for social problems involving stress. A vast number of studies confirm the positive effects of this activity on people's health. Nevertheless, little is known about the influence of winter forest bathing when conducted in an environment with snow cover on the ground and trees. Thus, a crossover experiment was designed in this study, with the participation of twenty-two healthy university students from Finland. During the experiment, a short exposition by a forest environment or landscape with buildings (as a control) was applied. Participants self-reported their psychological relaxation before and after the exposition, and the results were analyzed and compared. The mood, emotions, restorativeness, and subjective vitality were recorded as indices reflecting the psychological relaxation effect. The negative mood indices decreased significantly after exposition by the snow-covered environment, but the positive 'vigor' indices did not increase or decrease significantly. The level of negative emotions increased after the exposition with the control environment. Likewise, positive emotions decreased after the interaction with the control. Restorativeness was significantly increased after the exposition by the experimental forest but decreased after the viewing of the control buildings. The size of the effect in terms of restorativeness was the highest in this experiment. The subjective vitality was lowered as affected by the control, but it did not increase or decrease after the exposition with the experimental forest. There is probably an effect from the slight interruption in the process from the influence of the forest greens on participants because their vigor and vitality did not increase after the exposition with this environment in the study. However, snow might influence the participants as a calming and emotion-lowering component of the environment, but this idea needs to be further explored with the involvement of participants from other countries who would be viewing forest environments with snow cover and whose psychological relaxation could be measured.

"Psychological well-being in the winter forest: A pilot study", Mendeley Data, V1, doi: 10.17632/jprp23jn68.1.

**Funding:** The publication costs of this article were covered by the Forest Department of the Warsaw University of Life Sciences (SGGW) and from the Department of General Pedagogy at the University of Warmia and Mazury in Olsztyn.

**Competing interests:** The authors have declared that no competing interests exist.

## Introduction

Forest bathing (taking in the forest atmosphere, or Shinrin-Yoku) is a recreational outdoor activity conducted in a forest environment to induce a restorative experience or reduce stress [1]. It is an important activity, and the participation of members of modern societies in this type of activity might have promising effects on their health, especially regarding problems with a stress [2, 3]. Forest bathing as conducted in an organized way or performed by the subjects on their own might have a salutary influence on these problems [3–8]. This activity also has therapeutic potential for treating mental health problems, such as depression, posttraumatic stress disorder or schizophrenia [9–13]. In addition, forest bathing has been reported to evoke anti-stress effects on psychological and physiological parameters [1, 8, 13–21]. Other research showed its positive effect on anti-cancer protein levels in patients during cancer therapy [22].

A great deal of research in the area of forest bathing and other forms of nature-related recreation indicate that these activities might induce a psychological relaxation effect [23–25]. This effect on subjects may be measured using self-reporting psychometric techniques, using concepts of mood, affect, restorativeness and vitality, such as the Profile of Mood States Questionnaire, Positive and Negative Affect Schedule questionnaire, Restorative Outcome Scale questionnaire, and Subjective Vitality Scale questionnaire [26–29]. Thus, the psychological relaxation effect may be defined as a measurable, positive influence on psychological health and relaxation, and it could be measured using the abovementioned questionnaires [30–37].

Forest bathing might also be conducted during the wintertime in the areas of the globe where four seasons occur, and where snow appears in the forest landscape. This winter condition also has a positive influence on the psychological relaxation of participants, just as forest bathing conducted during vegetative seasons did [25, 38–40]. The identification of the forest recreation effect (including forest bathing) on psychological relaxation is also important in areas where winter prevails for a significant part of the year, including snow cover. For example, in Finland, a country where the winter lasts from early October to the middle of May (seven and a half months of winter in some parts of the Finnish Lapland), the importance of winter forest recreation is high [41, 42]. Outdoor recreation during the winter in these regions is also important due to economic concerns [43]. The occurrence of snow in the Lapland is also common; hence, outdoor recreation conducted in landscapes with snow is important as well. Thus, knowing how forest bathing conducted in a winter landscape with snow cover might influence the psychological relaxation of subjects involved in this activity is important for many societies, including Finnish ones. This activity is crucial because the subjects involved in this activity might experience psychological relaxation [39], which is crucial for their actual psychological health and for health prevention [44], and it also might be interesting for the market [45]; forest bathing in snow might be promoted as a product in these regions. Therefore, many Finish entrepreneurs are interested in knowing how being in nature affects their clients, and they want to use this information to market their nature-related products [46].

In the literature, there is not much research available on the influence of forest bathing with snow covered trees on psychological relaxation. There are some studies in which participants indicated some preferences for photographs presenting landscapes with snow-covered trees [45], but their psychological relaxation was not measured. In other studies, snow was visible on the ground, but not on the trees; in addition, this environment had a desired, positive influence on psychological relaxation [39]. Moreover, there is strong evidence that winter forest bathing without snow has a positive effect on participants [25]. The previous research, in which psychological relaxation was measured under winter conditions, was conducted with an 'intense' form of control, i.e., an urban road environment with intense urban traffic. For this

reason, it will be more appropriate if a calmer, silent environment without greens and only with buildings is used as a control. Nevertheless, there are studies in which some 'restraining factors', such as the view of the urban buildings in the forest landscape matrix [47] or the use of a laptop during the recreational experience, can waste these experiences and have a negative influence on the participant response in comparison to the control. Snow that fell in the forest on the ground cover and on trees might be seen as a 'restraining factor', and this factor should be scientifically examined. It might influence the psychological relaxation of participants who view this landscape [39, 45].

Therefore, the aim of this study was to examine the influence of a winter forest landscape, with the ground and trees covered by snow, on the psychological relaxation of young adults. This investigation was conducted using a silent, calm control in the same winter environment; however, it was surrounded only by buildings in the landscape.

## Materials and methods

### Ethical statement

This study was ethically reviewed and approved by the Ethical Review Board at the University of Warmia and Mazury in Olsztyn. The number of the ethical statement is 06/2018. All the procedures were performed in this study in accordance with the ethical standards of the Polish Committee of Ethics in Science and with the 1964 Helsinki Declaration´s later amendments.

### Participants

Twenty-two undergraduate students of Finnish nationality from Häme University of Applied Sciences (11 women, 11 men) participated in this study, and their mean age (± SD) was 22.5 years (± 4.67). Participation in the study was voluntary, and the students confirmed their willingness to be involved in the research through a written consent form (Ethics Committee approval number: 06/2018). The participants were divided randomly into two groups, group A or group B, each of which consisted of 11 participants. They received some information from researchers about the experiment beforehand, but information about the expected results was only given to participants after the study. The gender of the study participants should be balanced between the group A and the group B. In this crossover study, the general female and male groups were the same in the A group and the B group overall. A statistical power analysis was conducted using G*Power 3.1.9.4 free software for Mac (Heinrich Hein University, Düsseldorf, Germany) [48]. The actual power (1–β error probability) was calculated at 0.775. A statistical test 'ANOVA: repeated measure, within factors' was used, and a power analysis 'Post hoc: Compute achieved power' was applied with an effect size of 0.25 and α error of probability of 0.05. The power of properly prepared experiments is 0.8 or higher; hence, the statistical power in this experiment was close to acceptable.

### Experimental stimuli

The participants were involved in four different types of activities in this study, and after each activity, psychometric questionnaires were administered. The first activity was held in a room environment before the participants went to see the forest environment (Pre: Forest), and it reflected the normal, current mental state of the participants. The room environment was a classroom in one of the Evo campus buildings at Häme University of Applied Sciences (southern Finland) (Fig 1A). The second activity involved viewing a forest environment in which the ground and trees were covered by snow (Post: Forest) (Fig 1B). During this exposition, the participants needed to take a 5-minute walk from the room on campus to the forest, in which

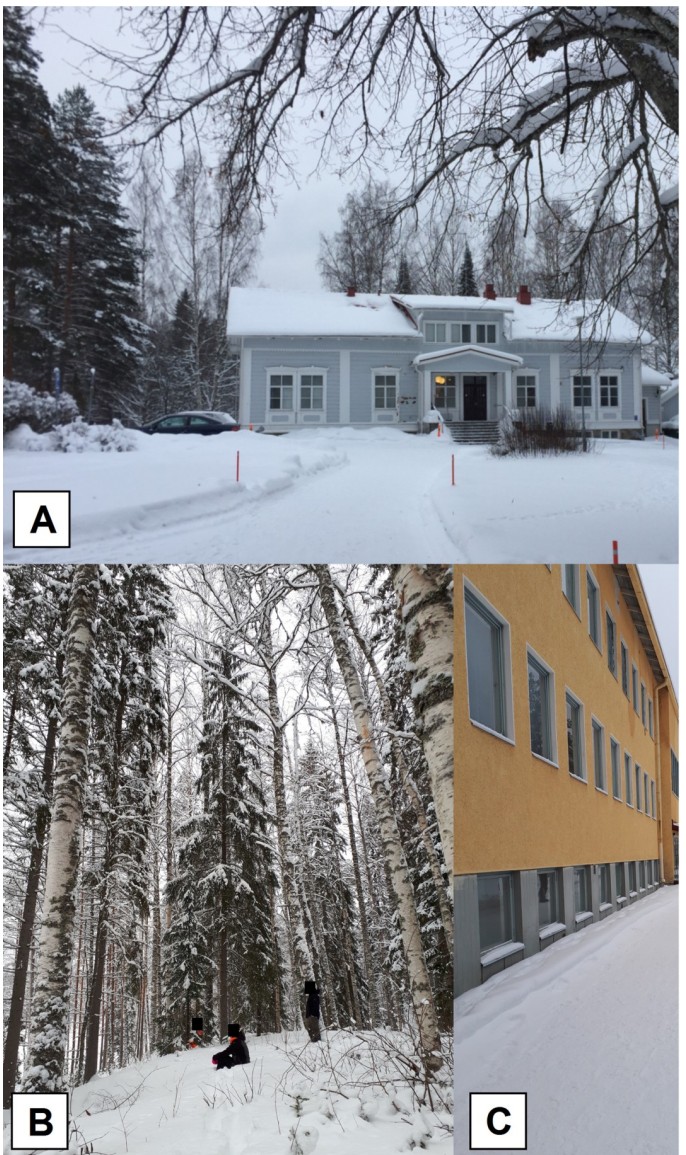

**Fig 1.** Photos showing the building with the room on the Evo campus (south Finland) for the pre-test (A), participants in standing or sitting positions (depending on the will of the participant) during the exposition by the forest environment with the ground and trees covered by snow (B) and during an exposition by a landscape with buildings (control) (C).

they stayed or sat for 15 minutes and contemplated the winter forest landscape. The students were allowed to move during the experiment because the temperature was below zero; however, talking, smoking or using electronic devices was forbidden. Additionally, the participants stood at some distance from each other (but in a line) so they did not have a chance to interrupt their neighbors (Fig 1B and 1C). The forest was composed of 80 to 108-year-old Norway spruces (close to 80%) and silver birches of the same age (close to 20%). The third activity (Pre: Buildings) was to stay in a room before viewing a landscape interrupted by buildings (a classroom on the Evo campus) (Fig 1C). The fourth activity was to view an outdoor winter landscape interrupted by buildings on the Evo campus (control environment), which was a two-minute walk from the room. The task of the participants during these activities was to

contemplate the landscape for 15 minutes in a sitting or standing position (Fig 1B and 1C). Talking, smoking or using electronic devices during this exposition was also forbidden. The order of participation in the activities of group A or B was reversed (crossover study) but staying in the room environment (activity 1 and 3) always occurred before exposition by each outdoor forest or building environment (activity 2 and 4).

## Procedure

The crossover experiment with the engaged students as participants was conducted on the 29[th] of January, at the Evo campus. On this day, two groups of students (A and B) participated in the experiment in random order. Each participant was involved in each measurement four times, i) after staying in the room environment before viewing the forest, ii) after viewing the forest environment, iii) after being in the room environment and before viewing the landscape with buildings, and iv) after viewing the landscape with buildings. The order depended on the participant affiliation with group A or B. The weather during the experiment was typical of January in southern Finland; the temperature was close to –7˚C (–7.23˚C ± 0.32), the sky was partially cloudy, there was a very slight wind (1.13 ± 0.71 m/s), the humidity was 94.25 ± 1.39% and the atmospheric pressure was 1000.49 ± 0.19 (Meteorological Station Hämeenlinna Lammi Evo, latitude: 61.21660, longitude: 25.13283). There was a 20-25-cm layer of snow cover on the ground and on the trees. The snow cover layer was similar in both experimental sites (in the forest and in front of the buildings). The climatic conditions were similar in both experimental sites, because there was no strong wind on that day, the differences were not felt by the respondents (oral report). During the measurement, the participants filled in questionnaires (after each exposition), while each questionnaire contained four psychometric tools: Profile of Mood States (POMS), Positive and Negative Affect Schedule (PANAS), Restorative Outcome Scale (ROS), and Subjective Vitality Scale (SVS). The respondents were asked to complete the questionnaire at each survey site (on-site).

## Measurements

POMS: The Profile of Mood States is a questionnaire used to measure six different mood states, namely tension-anxiety, depression-dejection, anger-hostility, fatigue, confusion, and vigor. The questionnaire is valid and commonly used [26]; a version with 65 items was applied in this research (with a 4-point Likert scale, from 0-not at all to 4-absolutely).

   PANAS: The Positive and Negative Affect Schedule (PANAS) is a questionnaire used to measure two types of emotional affect, positive and negative. This questionnaire, which contains 20 items, is valid and reliable [27], and a version with a 5-point Likert scale was used in our study.

   ROS: The Restorative Outcome Scale measures the restorative effect of each environment and contains six items. The scale is valid and reliable [49].

   SVS: The Subjective Vitality Scale contains four items and measures vitality. The scale is valid and reliable [29].

   All the questionnaires were given in English. The level of English proficiency among the Finnish students is very high, and in other studies, this method was used as well [50]. In case of any problems, the participants had a chance to ask bilingual academic teachers from the Evo campus to explain each confusing term in a questionnaire. The Cronbach's alphas (Table 1) were calculated to estimate the internal consistency and evaluate the usefulness of data for reasoning.

## Data and statistical analysis

Raw data from the questionnaires were used for the statistical analysis, and the mean values ± standard deviation were calculated for comparisons. Two-way repeated measures

**Table 1. Reliability of the experiment and number of items for each (sub) scale.**

| Scales and Subscales | Number of Items | Cronbach's Alpha |
|---|---|---|
| **POMS** | | |
| Tension | 9 | 0.832 |
| Depression-dejection | 15 | 0.924 |
| Anger-hostility | 12 | 0.899 |
| Vigor | 8 | 0.734 |
| Fatigue | 7 | 0.844 |
| Confusion | 7 | 0.809 |
| **PANAS** | | |
| Positive | 10 | 0.844 |
| Negative | 10 | 0.849 |
| **ROS** | 6 | 0.969 |
| **SVS** | 4 | 0.915 |

POMS: Profile of Mood States; PANAS: Positive and Negative Affect Schedule; ROS: Restorative Outcome Scale; and SVS: Subjective Vitality Scale.

ANOVAs (within-factors) were applied for the analysis, and the primary effects of the 'Condition', 'Time' and 'Condition × Time' Interaction were analyzed for the results of all the psychometric tools used here. After each ANOVA, a post hoc Tukey-Kramer multiple comparisons test was applied. The JMP 15 Trial for Mac with 'Full-factorial ANOVA Add-in' (SAS Institute Inc., Cary, NC, USA) was applied for ANOVAs and post hoc calculations. During the pre-test in groups A and B, there were a few missing values (for whole measurements of each participant), and these values were replaced with the values generated automatically by the Expectation Maximization (EM) technique in SPSS software.

## Results

### POMS

Two-way repeated measures ANOVA (within-factors) was used to analyze the effect of different conditions (forest environment vs. landscape with buildings), the effect of exposure to different environments (pre vs. post), and the interaction among them, on six different subscales of the POMS scale (six different mood states) (Table 2). Regarding the primary effects, the conditions had a significant effect on tension-anxiety, anger-hostility, and vigor (there was a non-significant but marginal effect on confusion). The primary effect of time had a significant effect on vigor and fatigue.

The results of Tukey-Kramer's multiple test comparisons showed that five POMS indicators were significantly lower (except for vigor, non-significant differences) after participant exposure to the forest environment than before (Forest: Pre vs. Post) (Table 3). After an exposition viewing of buildings (Buildings: Pre vs. Post), an increase was observed in the level of all five negative mood indicators (except for vigor, which is a positive mood indicator, which decreased, and depression-dejection, which was non-significant but had a high p-value (p = 0.071), and confusion, which was non-significant). By contrast, none of the POMS indicators differed significantly before the viewing of the forest environment or the landscape with buildings (Pre: Forest vs. Buildings). After the exposition, there were significant differences between forest and building observations in terms of POMS indicators; the values of all the negative indicators were significantly lower after exposure to the forest environment (fatigue, marginally non-significant p = 0.053) or higher for the positive vigor subscale (Post: Forest vs. Buildings).

**Table 2. Results of two-way repeated measures ANOVA for the Profile of Mood States (mood).**

| POMS | Primary effect | | | | | | | | Interaction | | | |
|---|---|---|---|---|---|---|---|---|---|---|---|---|
| | Condition: | | | | Time: | | | | Condition × Time | | | |
| | Buildings vs. Forest | | | | Pre vs. Post | | | | | | | |
| | F | P | | η² | F | P | | η² | F | P | | η² |
| Tension-anxiety | 9.32 | 0.006 | ** | 0.307 | 3.07 | 0.094 | | 0.128 | 43.74 | p<0.001 | *** | 0.676 |
| Depression-dejection | 2.34 | 0.141 | | 0.100 | 0.05 | 0.822 | | 0.002 | 17.92 | p<0.001 | *** | 0.460 |
| Anger-hostility | 10.93 | 0.003 | ** | 0.342 | 2.53 | 0.130 | | 0.107 | 19.95 | p<0.001 | *** | 0.487 |
| Vigor | 22.18 | p<0.001 | *** | 0.514 | 7.40 | 0.013 | * | 0.260 | 5.51 | 0.029 | * | 0.208 |
| Fatigue | 1.40 | 0.249 | | 0.063 | 21.90 | p<0.001 | *** | 0.511 | 8.18 | 0.009 | ** | 0.280 |
| Confusion | 3.76 | 0.066 | # | 0.152 | 26.91 | p<0.001 | *** | 0.562 | 23.59 | p<0.001 | *** | 0.529 |

*** $p < 0.001$,

**$p < 0.01$,

*$p < 0.05$, and

# $p < 0.1$ two-way repeated measures ANOVA.

## PANAS

A two-way repeated ANOVA of the PANAS data was applied, with the Condition and Time as effects and with the interaction of these two factors (Table 4). Regarding the primary effects, there was a significant effect of the Condition and Time on the positive aspect of PANAS. The Interaction Condition × Time was significant for both positive and negative aspects of PANAS.

**Table 3. Results of multiple comparison tests between forest and building (setting) and pre-post (exposure to the forest or the control) for the Profile of Mood States (mood).**

| | Forest | | | | | | Buildings | | | | | |
|---|---|---|---|---|---|---|---|---|---|---|---|---|
| | Pre | | Post | | | | Pre | | Post | | | |
| | Mean | S.D. | Mean | S.D. | P | | Mean | S.D. | Mean | S.D. | P | |
| Tension-anxiety | 1.20 | 0.55 | 0.58 | 0.5 | p<0.001 | *** | 1.04 | 0.29 | 1.38 | 0.44 | 0.023 | * |
| Depression-dejection | 0.76 | 0.45 | 0.43 | 0.58 | 0.035 | * | 0.56 | 0.28 | 0.86 | 0.57 | 0.071 | # |
| Anger-hostility | 0.62 | 0.40 | 0.41 | 0.5 | 0.003 | ** | 0.59 | 0.39 | 1.10 | 0.70 | 0.003 | ** |
| Vigor | 1.68 | 0.46 | 1.63 | 0.53 | 0.975 | | 1.54 | 0.28 | 1.11 | 0.46 | 0.009 | ** |
| Fatigue | 1.47 | 0.82 | 0.71 | 0.66 | 0.005 | ** | 1.31 | 0.42 | 1.16 | 0.64 | 0.053 | * |
| Confusion | 1.65 | 0.45 | 0.95 | 0.59 | p<0.001 | *** | 1.51 | 0.31 | 1.48 | 0.40 | 0.995 | |
| | Pre | | | | | | Post | | | | | |
| | Forest | | Buildings | | | | Forest | | Buildings | | | |
| | Mean | S.D. | Mean | S.D. | P | | Mean | S.D. | Mean | S.D. | P | |
| Tension-anxiety | 1.20 | 0.55 | 1.04 | 0.29 | 0.605 | | 0.58 | 0.5 | 1.38 | 0.44 | p<0.001 | *** |
| Depression-dejection | 0.76 | 0.45 | 0.56 | 0.28 | 0.844 | | 0.43 | 0.58 | 0.86 | 0.57 | 0.003 | ** |
| Anger-hostility | 0.62 | 0.40 | 0.59 | 0.39 | 0.994 | | 0.41 | 0.5 | 1.10 | 0.70 | p<0.001 | *** |
| Vigor | 1.68 | 0.46 | 1.54 | 0.28 | 0.980 | | 1.63 | 0.53 | 1.11 | 0.46 | p<0.001 | *** |
| Fatigue | 1.47 | 0.82 | 1.31 | 0.42 | 0.731 | | 0.71 | 0.66 | 1.16 | 0.64 | 0.053 | # |
| Confusion | 1.65 | 0.45 | 1.51 | 0.31 | 0.676 | | 0.95 | 0.59 | 1.48 | 0.40 | 0.001 | ** |

*** $p < 0.001$,

**$p < 0.01$,

*$p < 0.05$,

# $p < 0.1$ ANOVA-Tukey-Kramer.

**Table 4. Results of two-way repeated measures ANOVA for the Positive and Negative Affect Schedule (emotion).**

| PANAS | Primary effect | | | | | | | | Interaction | | | |
|---|---|---|---|---|---|---|---|---|---|---|---|---|
| | Condition: | | | | Time: | | | | Condition × Time | | | |
| | Building vs. Forest | | | | Pre vs. Post | | | | | | | |
| | F | P | | $\eta^2$ | F | P | | $\eta^2$ | F | P | | $\eta^2$ |
| Positive | 16.48 | 0.001 | ** | 0.440 | 26.00 | P < 0.001 | *** | 0.553 | 10.49 | 0.004 | ** | 0.333 |
| Negative | 0.28 | 0.601 | | 0.013 | 0.02 | 0.898 | | 0.001 | 18.76 | p < 0.001 | *** | 0.472 |

*** p < 0.001,

**p < 0.01 two-way repeated measures ANOVA.

The test of multiple comparisons (Table 5) showed that there were no differences between the pre-test (room environment before forest) and post-test for the forest environment for both negative and positive aspects of PANAS (Forest: Pre vs. Post), and there were significant differences for both aspects in between the room environment between viewing buildings and after viewing buildings (Buildings: Pre vs. Post); the positive aspect decreased and the negative aspect increased after the exposition from viewing buildings. There was a significant difference between the exposition by forest or buildings, and viewing buildings decreased the level of the positive aspect of PANAS (Post: Forest vs. Buildings).

## ROS

For the ROS two-way repeated measure, an ANOVA was conducted with two factors, Condition and Time, and their interaction was calculated as well (Table 6). The primary effect of the Condition was significant in this analysis, and the other primary effect of Time was not significant. The Interaction effect Condition × Time was significant.

The Tukey-Kramer's test of multiple comparisons (Table 7) showed that the ROS level significantly increased after participant exposure to the forest environment (Forest: Pre vs. Post). In addition, the ROS level decreased significantly after the exposition from the buildings (Buildings: Pre vs. Post). There was no significant difference between pre-tests before both experimental variants (Pre: Forest vs. Buildings). Furthermore, the ROS level increased significantly after the exposition by the forest environment and decreased after the exposition by buildings (Post: Forest vs. Buildings).

**Table 5. Results of multiple comparison tests between the forest and buildings (setting) and pre-post results (exposure to the forest or control) for the Positive and Negative Affect Schedule (emotion).**

| | Forest | | | | | Buildings | | | | | |
|---|---|---|---|---|---|---|---|---|---|---|---|
| | Pre | | Post | | | Pre | | Post | | | |
| | Mean | S.D. | Mean | S.D. | p | Mean | S.D. | Mean | S.D. | p | |
| Positive | 2.57 | 0.60 | 2.56 | 0.48 | 0.999 | 2.60 | 0.23 | 1.89 | 0.51 | p<0.001 | *** |
| Negative | 1.61 | 0.28 | 1.39 | 0.59 | 0.186 | 1.44 | 0.18 | 1.62 | 0.50 | 0.271 | |
| | Pre | | | | | Post | | | | | |
| | Forest | | Buildings | | | Forest | | Buildings | | | |
| | Mean | S.D. | Mean | S.D. | p | Mean | S.D. | Mean | S.D. | p | |
| Positive | 2.57 | 0.60 | 2.60 | 0.233 | 0.997 | 2.56 | 0.48 | 1.89 | 0.51 | p<0.001 | *** |
| Negative | 1.61 | 0.28 | 1.44 | 0.18 | 0.221 | 1.39 | 0.59 | 1.62 | 0.50 | 0.042 | * |

*** p < 0.001,

*p < 0.05, ANOVA-Tukey-Kramer.

**Table 6. Results of two-way repeated measures ANOVA for the Restorative Outcome Scale (subjective restorativeness).**

| ROS | Primary effect | | | | | | | Interaction | | | | |
|---|---|---|---|---|---|---|---|---|---|---|---|---|
| | Condition: | | | | Time: | | | Condition × Time | | | | |
| | Building vs. Forest | | | | Pre vs. Post | | | | | | | |
| | F | P | | η2 | F | P | η2 | F | P | | | η2 |
| | 35.53 | p < 0.001 | *** | 0.629 | 1.77 | 0.198 | 0.780 | 28.90 | p < 0.001 | *** | | 0.579 |

*** p < 0.001,

**p < 0.01 two-way repeated measures ANOVA.

## SVS

A two-way repeated measures ANOVA was conducted to compare the primary effects of the Condition and Time and the interaction between them for the SVS scores (Table 8).

There was a significant effect of the Condition and the Time on the SVS scores. The interaction was also significant.

The test of multiple comparisons (Table 9) showed that there was no difference between the pre-test (room environment before forest) and post-test (forest environment) results in the case of SVS (Forest: Pre vs. Post). A significant decrease was observed in the SVS values after participant exposure to the landscape with buildings (Buildings: Pre vs. Post). The room environments before viewing the forest environment did not differ from the room environment before viewing the landscape with buildings (Pre: Forest vs. Buildings). The viewing of the building had a significantly reduced SVS in comparison to the viewing of the forest environment (Post: Forest vs. Buildings).

## Discussion

### Mood states

Consistent with previous studies [2, 14, 18, 24, 25, 39, 40, 51–53], this study has confirmed that participation in a 15-minute forest bathing session has a positive effect on the mood states of participants. Overall, the snow covering the ground and the trees were not a restraining factor in these cases. The results of the other studies, in which the influence on participants of a forest environment with a thin (5 cm) cover of snow was examined, also showed that snow is not a strong restraining factor during the recreational experience, and this environment had a positive influence on the moods of participants [39]. Nevertheless, not every mood indicator was

**Table 7. Results of multiple comparison tests between forest and buildings (setting) and pre-post results (exposure to forest or buildings) for the Restorative Outcome Scale (subjective restorativeness).**

| | Forest | | | | | | Buildings | | | | | |
|---|---|---|---|---|---|---|---|---|---|---|---|---|
| | Pre | | Post | | | | Pre | | Post | | | |
| | Mean | S.D. | Mean | S.D. | P | | Mean | S.D. | Mean | S.D. | P | |
| ROS | 3.37 | 1.11 | 4.75 | 1.42 | 0.001 | ** | 3.37 | 0.79 | 1.67 | 0.68 | p < 0.001 | *** |
| | Pre | | | | | | Post | | | | | |
| | Forest | | Buildings | | | | Forest | | Buildings | | | |
| | Mean | S.D. | Mean | S.D. | P | | Mean | S.D. | Mean | S.D. | P | |
| ROS | 3.37 | 1.11 | 3.37 | 0.79 | 1.000 | | 4.75 | 1.42 | 1.67 | 0.68 | p < 0.001 | *** |

*** p < 0.001.

** p < 0.01. ANOVA-Tukey-Kramer.

**Table 8. Results of two-way repeated measures ANOVA for the Subjective Vitality Scale (subjective vitality).**

| SVS | Primary effect | | | | | | | | Interaction | | | |
|---|---|---|---|---|---|---|---|---|---|---|---|---|
| | Condition: | | | | Time: | | | | Condition × Time | | | |
| | Building vs. Forest | | | | Pre vs. Post | | | | | | | |
| | F | P | | η2 | F | P | | η2 | F | P | | η2 |
| | 5.55 | 0.028 | * | 0.209 | 17.27 | p < 0.001 | *** | 0.451 | 7.02 | 0.015 | * | 0.251 |

*** p < 0.001,

*p < 0.05 two-way repeated measures ANOVA.

easily changed by the forest environment in the present research, and the vigor level did not increase after the exposure to the forest environment (Table 2). By contrast, in research in which participants were involved in winter forest bathing, the level of vigor significantly increased after exposure to a forest environment without snow in the winter [25, 38]. It could be concluded that in some way, in this context, snow is a slight restraining factor because its presence stops the stimulation of participants in the areas responsible for vigor stimulation. The same observations were made in the study in which a thin snow cover occurred [39]. This means that, under certain circumstances, snow can suppress the feeling of vigor as it blocks the stimulating effect of green in the forest and the view of trees. This stimulating effect has been proven in other studies [25, 39]. These observations suggest that some greens that occurred in the forest environment (but were hidden under snow cover in these studies) might stimulate, in some specific way, the vigor of the participants. In addition, the other hypothesis is that the snow-covered environment might influence them not only by visual stimulation but also through a reduction in the concentration of volatiles in the air [54, 55], because in some other research, the vigor level increased after olfactory stimulation by etheric oil [56].

Notably, snow represents natural beauty, shelter (for animals and plants), and belongs to the nature in this area because of the four seasons. In addition, the white color might be perceived as an absence of aggressive colors [57]. Therefore, it seems reasonable that snow cover had no restraining effects on other mood variables. Nevertheless, the comparison of the POMS results with those from other regional studies conducted in forest environments was impossible, because this parameter has not been measured in this context in Finland before. The effect of calm control (environments with buildings not interrupted by entities, e.g., cars) on the POMS indices was significant, and the viewing of landscapes with buildings significantly decreased the mood of participants, but their confusion and depression-dejection were not significantly different.

**Table 9. Results of multiple comparison tests between forests and buildings (setting) and pre-post results (exposure to forest or buildings) for the Subjective Vitality Scale (subjective vitality).**

| | Forest | | | | | Buildings | | | | | |
|---|---|---|---|---|---|---|---|---|---|---|---|
| | Pre | | Post | | | Pre | | Post | | | |
| | Mean | S.D. | Mean | S.D. | P | Mean | S.D. | Mean | S.D. | P | |
| SVS | 4.42 | 0.78 | 4.36 | 0.97 | 0.999 | 4.39 | 0.70 | 3.57 | 0.91 | 0.002 | ** |
| | Pre | | | | | Post | | | | | |
| | Forest | | Buildings | | | Forest | | Buildings | | | |
| | Mean | S.D. | Mean | S.D. | P | Mean | S.D. | Mean | S.D. | P | |
| SVS | 4.42 | 0.78 | 4.39 | 0.70 | 0.999 | 4.36 | 0.97 | 3.57 | 0.91 | 0.011 | * |

** p < 0.01,

* p < 0.05, ANOVA-Tukey-Kramer.

In the study in which the POMS was measured during winter without snow cover, "the rough control" (control in the experiment in the city, related to i.e. car traffic), induced a negative effect on these two measures as well [25]; however, in the other study, when the snow partially covered the buildings, there was no significant effect on these parameters relative to "the rough control" [39]. The differences between the forest environment and the control environment are spectacular, and a good comparison is a pre-test, which was measured in the room environment and reflected the basic mood state of the participants.

## Emotions

The positive and negative effects of the PANAS changed in response to the forest or control environments, respectively, as confirmed in other studies [14, 25, 39, 52, 58]. Only the control environment had a significant influence on the positive and negative aspects of PANAS; the positive aspect decreased after participants viewed the landscape with buildings, but the negative aspect increased in this case. In other studies, with the use of rush control, the effect on PANAS was quite similar; the negative aspect increased, while the positive aspect decreased (but not significantly) after participants viewed the urban environment (street) during a winter without snow cover [25] and similarly under a thin snow cover [39]. In Tyrväinen et al. [3], the positive aspect of PANAS increased in the forest environment but decreased in the urban environment. The negative aspect of PANAS increased in an urban environment during the vegetation season. In the present research, the increase of these two aspects of PANAS in the forest environment was not significant in the winter with snow cover. In the available literature, there are also not many positions with information about the effect of the forest environment on the PANAS level among participants, because these indices are rarely measured.

## Restorativeness

The values of the Restorative Outcome Scale increased in the forest environment, similar to other studies [14, 25, 39, 52, 53, 58]. This psychological index is very sensitive at measuring the effects of forest environments on participants. In this study, the size of the effect was the highest among all the analyzed indices for interactions ($\eta^2 = 0.579$), and thus the effect of different environments on the ROS was high. In addition, forest with snow cover on the ground and on trees had a strong impact on this parameter, as strong as that in other studies [39], and the ROS has the largest effect of all the analyzed indices, but overall, the value of $\eta^2$ was lower in the cited study ($\eta^2 = 0.228$ for interaction). The reaction of the Finnish participants in the forest environment was previously measured with the ROS [58]. The ROS value in the control environment decreased, as in the rush control in the winter; in this study, it decreased to approximately 50% in comparison to the room environment, but in another study, this decrease was smaller in the rush control [39].

## Subjective vitality

The Subjective Vitality Scale is also a sensitive measure for indicating the effect of the forest environment; it detected this effect well here, as in other studies [3, 14, 21, 25, 39, 53, 58]. The size of the effect on the interaction was relatively small in comparison to other indices used in this research, and the differences were significant only for the forest environment in comparison to the landscape with buildings, and thus, the influence of the forest environment with snow cover had a relatively slight impact on the SVS. This could be a similar situation to the subscale of POMS on vigor; snow is probably a slightly 'restraining factor', which halted the increase in vigor and subjective vitality, which needs to be further explored. The calm control significantly decreased the level of SVS, but it might be specific for the Finnish population

because in the rush control in the experiment with snow cover, this effect did not occur [39], but this effect needs to be elucidated in further investigations.

In a previous study [25], the vitality was increased as well with Polish participants, in the country with a shorter winter. Populations of northern countries feel winter differently because it is longer, darker and often makes them depressed [59, 60]. For the Finnish people, being outside in the cold forest in the wintertime may also appear as a life-threatening situation, which could also affect the other indices (mood, emotions, and restorativeness).

## Limitations

In order to improve the statistical power, it is proposed to perform further studies in which a larger number of respondents and also non-students will participate. In this study, the number of participants was relatively small, but the power of the test was statistically acceptable. In subsequent studies, it is proposed to perform a power analysis of the experiment before collecting data, to obtain information about how large a group of subjects will be sufficient.

In future studies involving non-student participants, the socio-demographic variables should be considered and balanced for each group. Additionally, the assessment of the contact and attitude of the subjects towards the snow environment should also be measured, for example, the level of pleasure induced by snow in the subjects and the experience of sports in the snow may be variables and covariates in this study.

In the current study, a pilot study was carried out on the impact of the snow environment on the subjects. The approach here planned was to expose the subjects to 15 minutes of exposition. However, this is only one type of activity that can be considered, true outdoor recreation often lasts longer and provides a variety of stimuli, what should be considered in future research. It can be imagined that in future studies the experimental group would participate in a two-hour walk in the snow in the forest, or be involved in sports activities in this environment. As a result, it will be possible to obtain results on a more natural form of activity in a forest environment with snow cover.

Another limitation of this work is the use of research in the winter season, but a good reference point would be to perform additional experiments in the growing season. Such an experimental setup will be implemented in future research. We were interested in the winter aspect, mainly because this period in quite a large part of Phenoscandia lasts extremely long, and the long periods of snow cover are inextricably linked with this region of Europe. Hence, the results of the research may have very important regarding implementations for the inhabitants of Finland.

Another limitation of the research was the differentiation of the way of psychological relaxation—the subjects could rest while sitting or standing and looking at the forest. This variable (sitting or standing during relaxation in the forest) was not controlled in the experiment. Another limitation was the fact that the climatic conditions of the environment were not accurately measured in each experimental site (in the forest and in front of the buildings). Both of these limitations will be considered and eliminated in future experiments.

The main limitation of these studies is the low power of the experiment resulting from the small number of subjects (22 people), but to verify the correctness of the research, we calculated the power of the experiment in this study and it is statistically acceptable. In other studies [61, 62] the number of participants was 12–15, which has been considered sufficient to draw preliminary conclusions. Research of this type is characterized by its specificity, which means that they are not conducted in large samples, additionally, coefficients are used to ensure repeatability of the results, i.e. Cronbach's alpha.

## Conclusions

This study examined the effect of winter forest bathing in a snow-covered environment, with snow on the ground and on the trees compared with a calm landscape with buildings (no cars, no traffic) used as a control on the psychological relaxation of young Finnish students at HAMK University. The results showed that the level of negative mood indicators (subscales of POMS scale) among the participants primarily significantly decreased after their exposure to a snow-covered forest environment, and a positive indicator of 'vigor' did not change this time. The level of positive emotions (PANAS Positive) decreased after participants viewed a calm control with buildings, and in this environment, the level of negative emotions (PANAS Negative) increased. In the forest environment, these two indicators did not increase significantly in comparison to a pre-test (normal, expected state of participants). The restorativeness of the snow-covered environment was high, the ROS indicator increased significantly, and the effect size was the highest of all the psychological indicators included in this study. The calm control significantly decreased the level of ROS at the same level as the rush control did in other studies. The subjective vitality (SVS) did not increase after participant exposure to the snow-covered forest environment but decreased after their exposure to the calm control. In other studies, this indicator increased after exposure to a forest environment in a vegetative state. It is possible that the snow cover is a slightly restraining factor in the environment, which halted the impact of the green on the vigor and vitality of participants, but this idea needs to be elucidated in further investigations.

## Author Contributions

**Conceptualization:** Ernest Bielinis, Emilia Janeczko, Norimasa Takayama, Anna Zawadzka, Alicja Słupska, Sławomir Piętka, Maija Lipponen, Lidia Bielinis.

**Data curation:** Ernest Bielinis, Emilia Janeczko, Lidia Bielinis.

**Formal analysis:** Ernest Bielinis, Emilia Janeczko, Norimasa Takayama, Anna Zawadzka, Alicja Słupska, Sławomir Piętka, Lidia Bielinis.

**Funding acquisition:** Lidia Bielinis.

**Investigation:** Ernest Bielinis, Norimasa Takayama, Maija Lipponen, Lidia Bielinis.

**Methodology:** Ernest Bielinis, Emilia Janeczko, Norimasa Takayama, Lidia Bielinis.

**Project administration:** Anna Zawadzka, Alicja Słupska.

**Resources:** Lidia Bielinis.

**Supervision:** Norimasa Takayama, Anna Zawadzka, Alicja Słupska, Sławomir Piętka, Maija Lipponen, Lidia Bielinis.

**Validation:** Maija Lipponen.

**Visualization:** Sławomir Piętka.

**Writing – original draft:** Ernest Bielinis.

**Writing – review & editing:** Emilia Janeczko, Norimasa Takayama, Anna Zawadzka, Alicja Słupska, Sławomir Piętka, Maija Lipponen, Lidia Bielinis.

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
