## [Decision Letter · Decision Letter 0]

14 Jul 2020

PONE-D-20-06816

The Effects of Viewing Winter Forest Landscape With Ground and Trees Covered By Snow on the Psychological Relaxation of Young Finnish Adults

PLOS ONE

Dear Dr. Bielinis,

Thank you for submitting your manuscript to PLOS ONE. After careful consideration, we feel that it has merit but does not fully meet PLOS ONE’s publication criteria as it currently stands. Therefore, we invite you to submit a revised version of the manuscript that addresses the points raised during the review process.

We look forward to receiving your revised manuscript.

Kind regards,

Marco Innamorati

Academic Editor

PLOS ONE

Journal Requirements:

2. Please provide additional details regarding participant consent. In the Methods section, please ensure that you have specified what type of consent you obtained (for instance, written or verbal) and whether the ethics committee approved this consent procedure. If verbal consent was obtained please state why it was not possible to obtain written consent and how verbal consent was recorded. If your study included minors, state whether you obtained consent from parents or guardians.

5. Please amend either the title on the online submission form (via Edit Submission) or the title in the manuscript so that they are identical.

7. We note that Figure [B & C] includes an image of a [patient / participant / in the study]. 

Reviewers' comments:

Reviewer's Responses to Questions

**Comments to the Author**

1. Is the manuscript technically sound, and do the data support the conclusions?

Reviewer #1: No

Reviewer #2: Partly

2. Has the statistical analysis been performed appropriately and rigorously? 

Reviewer #1: Yes

Reviewer #2: Yes

3. Have the authors made all data underlying the findings in their manuscript fully available?

Reviewer #1: No

Reviewer #2: Yes

4. Is the manuscript presented in an intelligible fashion and written in standard English?

Reviewer #1: No

Reviewer #2: Yes

5. Review Comments to the Author

Reviewer #1: Thank you for the opportunity to review the manuscript titled: “The Effects of Viewing Winter Forest Landscape With Ground and Trees Covered By Snow on the Psychological Relaxation of Young Finnish Adults”. The manuscript under review deals with the relationships between contact with nature and psychological wellbeing outcomes, such as emotion, restorativeness and subjective vitality. The topic of the manuscript would seem to be relevant and deserves attention. Nevertheless, I think a major revision of the manuscript is needed. I strongly recommend to test such relationships through a second study collecting a bigger sample and including non-student participants. The current sample is very small and includes only student participants. It is a fine pilot study. In a second study, an a priori statistical power analysis should be carried out before the collecting data. Participants gender needs to be balanced between experimental and control groups. In the current manuscript, it is not clear the composition of each group concerning the gender variable. A second study involving non-student participants would need to take into account socio-demographic variables and such variables have to be balanced for each group. Even though the study has been carried out in Finland a measure of confidence with snowy environments (e.g., pleasantness of snowy environments, skilled skier) should be assessed in order to control such a variable as a covariate. Authors mention such an aspect, in the discussion section, when referring to Finland winter and this variable could address this issue. In the discussion more can be done to provide suggestions concerning applied interventions as well as future researches.

Minor comments:

The current version of the manuscript needs to be edited because there are some typos. For example, p.12 row 250, in the following statement: “with the Condition as … and Time” a word is missing. There are other typos which need to be edited throughout the manuscript.

Authors should replace “males” and “females” with “women” and “men”. Even though females and males have been widely used in the past, women and men are more suitable terms for current academic writing because it takes out gendered stereotypes.

Reviewer #2: Authors highlight an interesting approach. Notwithstanding this aspect, the paper shows several limitations. Firstly, to investigate the role of snow in positive effect on psychological aspects such as relaxation they should complete the experiment by introducing another condition about a landscape without snow. Secondly, the paper miss several bibliographic references (i.e., there are numerous phrases missing of reference). The major concern is about the lack of power due to the small sample size

6. PLOS authors have the option to publish the peer review history of their article (what does this mean?). If published, this will include your full peer review and any attached files.

Reviewer #1: No

Reviewer #2: No

---

## [Author Response · Author response to Decision Letter 0]

15 Sep 2020

Response to Editors’s:

Ad. 1. Done

Ad 2. Additional details were added.

Ad. 3. Data have private information about participants and cannot be spread online.

Ad. 4. Done

Ad. 5. Done

Ad. 6. Done

Ad. 7. We shaded faces and elements responsible for recognition in participants on the photographs, so as we think, now this photograph can be officially publish. 

Review Comments to the Author

Reviewer #1: Thank you for the opportunity to review the manuscript titled: “The Effects of Viewing Winter Forest Landscape With Ground and Trees Covered By Snow on the Psychological Relaxation of Young Finnish Adults”. The manuscript under review deals with the relationships between contact with nature and psychological wellbeing outcomes, such as emotion, restorativeness and subjective vitality. The topic of the manuscript would seem to be relevant and deserves attention.

Thank you to reviewer # 1 for this comment. We are grateful that Reviewer # 1 appreciated our work and our topic selection.

Nevertheless, I think a major revision of the manuscript is needed. I strongly recommend to test such relationships through a second study collecting a bigger sample and including non-student participants. The current sample is very small and includes only student participants. It is a fine pilot study

Thank you to reviewer # 1 for this comment. The presented studies are the result of preliminary studies conducted in Finland in winter. These are, of course, a pilot study, and to underline this we decided to add the statement "pilot study" in the title of this manuscript.

Since this is a preliminary study, there are some limitations that we take into account, so we have added a "limitations" section to this manuscript. It would be difficult to carry out additional research in the near future, due to the time of the year, which will be in a few months (winter) and the constraints related to the pandemic, and the topic is important and in our opinion, the latest report on the impact of snow on the respondents deserves publication.

So if the reviewer kindly allows, we will highlight in this manuscript the fact that this is a pilot study (by adding the statement "pilot study" in the title) and refer to the individual improvements proposed by the reviewer, these references will be included in the "limitations" chapter. In our opinion, thanks to this solution, it will be possible to publish the manuscript with these results, but it will be known to the reader what the limitations of the research are. We believe that this will be a compromise between the requirements of Reviewer # 1 and our team's ability to conduct another study at this point.

In the "limitations" chapter we added:

In order to improve the statistical power, it is proposed to perform further studies in which a larger number of respondents and also non-students will participate. In this study, the number of participants was relatively small, but the power of the test was statistically acceptable.

In a second study, an a priori statistical power analysis should be carried out before the collecting data. 

Thank you to reviewer # 1 for this comment. In the "limitations" chapter we added:

In subsequent studies, it is proposed to perform a power analysis of the experiment before collecting data, to obtain information about how large a group of subjects will be sufficient.

Participants gender needs to be balanced between experimental and control groups. In the current manuscript, it is not clear the composition of each group concerning the gender variable. 

Thank you to reviewer # 1 for this comment. In the chapter "materials and methods" we added:

The gender of the study participants should be balanced between the experimental group and the control group. In this crossover study, the general female and male groups were the same in the control group and the experimental group overall.

A second study involving non-student participants would need to take into account socio-demographic variables and such variables have to be balanced for each group.

Thank you to reviewer # 1 for this comment. In the "limitations" chapter we added:

In future studies involving non-student participants, the socio-demographic variables should be considered and balanced for each group.

Even though the study has been carried out in Finland a measure of confidence with snowy environments (e.g., pleasantness of snowy environments, skilled skier) should be assessed in order to control such a variable as a covariate. Authors mention such an aspect, in the discussion section, when referring to Finland winter and this variable could address this issue.

Thank you to reviewer # 1 for this comment. In the "limitations" chapter we added:

In future studies, the assessment of the contact and attitude of the subjects towards the snow environment should also be measured, for example, the level of pleasure induced by snow in the subjects and the experience of sports in the snow may be variables and covariates in this study.

In the discussion more can be done to provide suggestions concerning applied interventions as well as future researches.

Thank you to reviewer # 1 for this comment. In the "limitations" chapter we added:

In the current study, a pilot study was carried out on the impact of the snow environment on the subjects. The approach here planned was to expose the subjects to 15 minutes of stimulation. However, this is only one type of activity that can be considered, true outdoor recreation often lasts longer and provides a variety of stimuli, and in future research these natural activities should be considered as experimental and tested stimulation. It can be imagined that in future studies the experimental group would participate in a two-hour walk in the snow in the forest, or be involved in sports activities in this environment. As a result, a more natural form of activity could be tested in a forest environment with snow cover.

Minor comments:

The current version of the manuscript needs to be edited because there are some typos. For example, p.12 row 250, in the following statement: “with the Condition as … and Time” a word is missing. There are other typos which need to be edited throughout the manuscript.

Authors should replace “males” and “females” with “women” and “men”. Even though females and males have been widely used in the past, women and men are more suitable terms for current academic writing because it takes out gendered stereotypes.

Thanks to reviewer # 1 for these comments. We included these minor changes to the manuscript.

Reviewer #2: Authors highlight an interesting approach. 

Thank you to reviewer # 2 for this comment. We are grateful that Reviewer # 1 appreciated our work and our topic selection.

Notwithstanding this aspect, the paper shows several limitations. 

Firstly, to investigate the role of snow in positive effect on psychological aspects such as relaxation they should complete the experiment by introducing another condition about a landscape without snow.

Thank you to reviewer # 2 for this comment. As this is a pilot study, the number of options has been reduced as much as possible, we will include this approach in future studies. If the reviewer allows, we will present this approach as a limitation of our research, while proposing future research. At the same time, we would like to draw attention to the fact that in many studies the type of experiment was presented as control + experimental variant, then the influence of forest areas during the growing season on the participants of the experiment was examined, and in this type of research there were no additional variants or additional control. We adopted a similar course of action in our experiment.

We have added the following in the "limitations" section:

Another limitation of this work is the use of research in the winter season, but a good reference point would be to perform additional experiments in the growing season. Such an experimental setup will be implemented in future research. We were interested in the winter aspect, mainly because this period in quite a large part of Phenoscandia lasts extremely long, and the long periods of snow cover are inextricably linked with this region of Europe. Hence, the results of the research may have very important implementations for the inhabitants of Finland

Secondly, the paper miss several bibliographic references (i.e., there are numerous phrases missing of reference). 

Thank you to reviewer # 2 for this comment. After careful verification of the text, we added the missing literature items.

The major concern is about the lack of power due to the small sample size.

Thank you to reviewer # 2 for this comment. This is obviously a limitation of this research. We have added the following in the "limitations" section:

The main limitation of these studies is the low power of the experiment resulting from the small number of subjects (22 people), but to control this fact, we calculated the power of the experiment in this study and it is statistically acceptable. In other studies (including Takayma et al.), The number of participants was 12-15, which was sufficient to draw meaningful conclusions. Research of this type is characterized by its specificity, which means that they are not conducted in large samples, additionally, coefficients are used to ensure repeatability of the results, i.e. Cronbach's alpha.

---

## [Decision Letter · Decision Letter 1]

2 Nov 2020

PONE-D-20-06816R1

The Effects of Viewing a Winter Forest Landscape with the Ground and Trees Covered in Snow on the Psychological Relaxation of Young Finnish Adults: A Pilot Study

PLOS ONE

Dear Dr. Bielinis,

Thank you for submitting your manuscript to PLOS ONE. After careful consideration, we feel that it has merit but does not fully meet PLOS ONE’s publication criteria as it currently stands. Therefore, we invite you to submit a revised version of the manuscript that addresses the points raised during the review process.

We look forward to receiving your revised manuscript.

Kind regards,

Marco Innamorati

Academic Editor

PLOS ONE

Reviewers' comments:

Reviewer's Responses to Questions

**Comments to the Author**

1. If the authors have adequately addressed your comments raised in a previous round of review and you feel that this manuscript is now acceptable for publication, you may indicate that here to bypass the “Comments to the Author” section, enter your conflict of interest statement in the “Confidential to Editor” section, and submit your "Accept" recommendation.

Reviewer #1: (No Response)

Reviewer #2: (No Response)

2. Is the manuscript technically sound, and do the data support the conclusions?

Reviewer #1: (No Response)

Reviewer #2: Partly

3. Has the statistical analysis been performed appropriately and rigorously? 

Reviewer #1: (No Response)

Reviewer #2: Yes

4. Have the authors made all data underlying the findings in their manuscript fully available?

Reviewer #1: (No Response)

Reviewer #2: No

5. Is the manuscript presented in an intelligible fashion and written in standard English?

Reviewer #1: (No Response)

Reviewer #2: No

6. Review Comments to the Author

Reviewer #1: In the current version of the manuscript, Authors recognize several limitations of the research and they use a more cautious title. Nevertheless, there is a statements that should be dampened. The statement, "In other studies [56,57] the number of participants was 12-15, which was sufficient to draw meaningful conclusions" should be reworded in the following way: "In other studies [56,57] the number of participants was 12-15, which HAS BEEN CONSIDERED sufficient to draw PRELIMINARY conclusions." Previous underpowered studies are not a justification to carry out future studies underpowered.

Even though the manuscript report limitations, Authors recognize such limitations in the discussion section and provide some suggestions for future studies. Moreover, the topic is somewhat original and might stimulate future research. Thus, in my view, the manuscript could be accepted as a pilot study after this round of minor revision by which Authors dampen the statement above reported.

Reviewer #2: Part of my concerns have been adressed by authors. Unfortunately, there are important issues that continue to be present in the paper like uncorrect use of some terms and not clear statements that have to be corrected or clarified. In general, the study should be described with greater methodological rigor.

Examples are:

-in my opinion, the term "stimulation" used to describe experimental conditions should be substituted with "exposition"

- you define the sample as "healthy" but this statement is not supported with any test (line 128)

- if groups are equal in composition and experimental conditions they received, why did you define two groups as experimental and control (e.g., line 138)? In statistical analyses section you use a within design. You should clarify this aspect

- In lines 73-74, bibliographic references are necessary. It is a crucial point of the paper because you operatively define the "psychological relaxation effect".

- in both buildings and forest conditions subjects could stay stayed or sat. Given the importance of these two condition on "psychological relaxation effect", this aspect should to be controlled.

-"sample size of 0.25" should be corrected with "effect size of 0.25" (line 143)

-lines 136-138 are not clear

- you should clarify if the climate features (e.g., temperature) are the same in "forest" and "building" conditions. Indeed, the snow was a prominent feature of forest condition, but you didn't report the state of snow in building condition.

-lines 349-354 are not clear, you should explain better these concepts and support them with bibliografic references

-in line 371 what do you mean with "rough control"?

-lines 480-481 should be clarified

7. PLOS authors have the option to publish the peer review history of their article (what does this mean?). If published, this will include your full peer review and any attached files.

Reviewer #1: No

Reviewer #2: No

---

## [Author Response · Author response to Decision Letter 1]

10 Nov 2020

Reviewer #1: 

We thank the reviewer for these comments, we hope our responses to these comments are sufficient, and appreciate the reviewer's contribution to improving the quality of this work.

In the current version of the manuscript, Authors recognize several limitations of the research and they use a more cautious title. Nevertheless, there is a statements that should be dampened. The statement, "In other studies [56,57] the number of participants was 12-15, which was sufficient to draw meaningful conclusions" should be reworded in the following way: "In other studies [56,57] the number of participants was 12-15, which HAS BEEN CONSIDERED sufficient to draw PRELIMINARY conclusions." Previous underpowered studies are not a justification to carry out future studies underpowered.

Thanks to reviewer # 1 for this comment, of course we made some corrections.

Even though the manuscript report limitations, Authors recognize such limitations in the discussion section and provide some suggestions for future studies. Moreover, the topic is somewhat original and might stimulate future research. Thus, in my view, the manuscript could be accepted as a pilot study after this round of minor revision by which Authors dampen the statement above reported.

Thank you to the reviewer for appreciating this topic and our corrections.

Reviewer #2: 

Part of my concerns have been adressed by authors. Unfortunately, there are important issues that continue to be present in the paper like uncorrect use of some terms and not clear statements that have to be corrected or clarified. In general, the study should be described with greater methodological rigor.

We thank the reviewer for these comments, we hope our responses to these comments are sufficient, and appreciate the reviewer's contribution to improving the quality of this work.

Examples are:

-in my opinion, the term "stimulation" used to describe experimental conditions should be substituted with "exposition"

Thank you for this comment, we've made some corrections.

- you define the sample as "healthy" but this statement is not supported with any test (line 128)

Thank you for this comment, we have removed this adjective.

- if groups are equal in composition and experimental conditions they received, why did you define two groups as experimental and control (e.g., line 138)? In statistical analyses section you use a within design. You should clarify this aspect

We thank the reviewer for this comment. Of course we meant group A and group B. It was a crossover study (control and experimental groups were exchanging, so this is the same as group A and group B). We have supplemented this in the text.

- In lines 73-74, bibliographic references are necessary. It is a crucial point of the paper because you operatively define the "psychological relaxation effect".

We thank the reviewer for this comment. We have added needed literature here.

- in both buildings and forest conditions subjects could stay stayed or sat. Given the importance of these two condition on "psychological relaxation effect", this aspect should to be controlled.

We thank the reviewer for this comment. Unfortunately, we have not controlled this effect, so we consider it a research limitation (if the reviewer kindly allows). We added an explanation:

“Another limitation of the research was the differentiation of the way of psychological relaxation - the subjects could rest while sitting or standing and look at the forest. However, this variable, ie sitting or standing, was not controlled in the experiment.”

\\

-"sample size of 0.25" should be corrected with "effect size of 0.25" (line 143)

We thank the reviewer for this comment. Yes, we have corrected it.

-lines 136-138 are not clear

We thank the reviewer for this comment. We clarified it, it was about groups A and B ( which are interchangeably control and corossover study group).

- you should clarify if the climate features (e.g., temperature) are the same in "forest" and "building" conditions. Indeed, the snow was a prominent feature of forest condition, but you didn't report the state of snow in building condition.

We thank the reviewer for this comment. We have added the following text:

„The snow cover layer was similar in both experimental sites (in the forest and in front of the buildings). The climatic conditions were similar in both experimental sites, because there was no strong wind on that day, the differences were not felt by the respondents (oral report).”

and in limitations:

“Another limitation may be the fact that the climatic conditions of the environment were not accurately measured in each experimental site (in the forest and in front of buildings). Both of these limitations will be considered and eliminated in the next experiment.”

-lines 349-354 are not clear, you should explain better these concepts and support them with bibliografic references

Thank you to recent for this comment. We have added the following comment:

“This means that we believe that, under certain circumstances, snow can suppress the feeling of vigor as it blocks the stimulating effect of green in the forest and the view of trees. This stimulating effect has been proven in other studies”

-in line 371 what do you mean with "rough control"?

Thank you to recent for this comment. We've added the following text, and used this statement in quotation marks. 

“(control in an experiment in the city, related to, inter alia, car traffic).”

-lines 480-481 should be clarified

Thank you to recent for this comment. We dropped that sentence - it was deleted.

---

## [Decision Letter · Decision Letter 2]

11 Dec 2020

PONE-D-20-06816R2

The Effects of Viewing a Winter Forest Landscape with the Ground and Trees Covered in Snow on the Psychological Relaxation of Young Finnish Adults: A Pilot Study

PLOS ONE

Dear Dr. Bielinis,

Thank you for submitting your manuscript to PLOS ONE. After careful consideration, we feel that it has merit but does not fully meet PLOS ONE’s publication criteria as it currently stands. Therefore, we invite you to submit a revised version of the manuscript that addresses the points raised during the review process.

We look forward to receiving your revised manuscript.

Kind regards,

Marco Innamorati

Academic Editor

PLOS ONE

Reviewers' comments:

Reviewer's Responses to Questions

**Comments to the Author**

1. If the authors have adequately addressed your comments raised in a previous round of review and you feel that this manuscript is now acceptable for publication, you may indicate that here to bypass the “Comments to the Author” section, enter your conflict of interest statement in the “Confidential to Editor” section, and submit your "Accept" recommendation.

Reviewer #1: All comments have been addressed

Reviewer #2: All comments have been addressed

Reviewer #3: (No Response)

2. Is the manuscript technically sound, and do the data support the conclusions?

Reviewer #1: Yes

Reviewer #2: Partly

Reviewer #3: Partly

3. Has the statistical analysis been performed appropriately and rigorously? 

Reviewer #1: Yes

Reviewer #2: Yes

Reviewer #3: Yes

4. Have the authors made all data underlying the findings in their manuscript fully available?

Reviewer #1: (No Response)

Reviewer #2: Yes

Reviewer #3: Yes

5. Is the manuscript presented in an intelligible fashion and written in standard English?

Reviewer #1: Yes

Reviewer #2: No

Reviewer #3: Yes

6. Review Comments to the Author

Reviewer #1: (No Response)

Reviewer #2: Comments and concerns have been adressed. Limitations have been properly highlighted. I suggest to authors a further English check.

Reviewer #3: - It has been already said that the experimental subjects were randomly assigned to the two groups (line 133), there is no need to repeat it in line 179

- In lines 359-36: "the other strong hypothesis is that the snow covered environment might influence them not only by visual stimulation but also through a reduction in the concentration of volatiles in the air". This aspect is not directly measured, so it may be more appropriate to re-evaluate the term "strong hypothesis" used in the paper

7. PLOS authors have the option to publish the peer review history of their article (what does this mean?). If published, this will include your full peer review and any attached files.

Reviewer #1: No

Reviewer #2: No

Reviewer #3: No

---

## [Author Response · Author response to Decision Letter 2]

15 Dec 2020

Response to reviewers:

Reviewer #1: (No Response)

Reply:

We thank the reviewer for this comment.

Reviewer #2: Comments and concerns have been addressed. Limitations have been properly highlighted. I suggest to authors a further English check.

Reply:

We would like to thank the reviewer for this comment, we have made language corrections as a result of an additional language check two times by qualified proofreader and also proofreading services.

Reviewer #3: 

- It has been already said that the experimental subjects were randomly assigned to the two groups (line 133), there is no need to repeat it in line 179

Reply: 

We thank the reviewer for this comment, we have removed the repetitive fragment of the text.

- In lines 359-36: "the other strong hypothesis is that the snow covered environment might influence them not only by visual stimulation but also through a reduction in the concentration of volatiles in the air". This aspect is not directly measured, so it may be more appropriate to re-evaluate the term "strong hypothesis" used in the paper

Reply:

We thank the reviewer for this comment, we reevaluated the meaning of this sequence, removed the word "strong".

---

## [Decision Letter · Decision Letter 3]

17 Dec 2020

The Effects of Viewing a Winter Forest Landscape with the Ground and Trees Covered in Snow on the Psychological Relaxation of Young Finnish Adults: A Pilot Study

PONE-D-20-06816R3

Dear Dr. Bielinis,

We’re pleased to inform you that your manuscript has been judged scientifically suitable for publication and will be formally accepted for publication once it meets all outstanding technical requirements.

Kind regards,

Marco Innamorati

Academic Editor

PLOS ONE

Additional Editor Comments (optional):

Reviewers' comments:

Reviewer's Responses to Questions

**Comments to the Author**

1. If the authors have adequately addressed your comments raised in a previous round of review and you feel that this manuscript is now acceptable for publication, you may indicate that here to bypass the “Comments to the Author” section, enter your conflict of interest statement in the “Confidential to Editor” section, and submit your "Accept" recommendation.

Reviewer #3: All comments have been addressed

2. Is the manuscript technically sound, and do the data support the conclusions?

Reviewer #3: Yes

3. Has the statistical analysis been performed appropriately and rigorously? 

Reviewer #3: Yes

4. Have the authors made all data underlying the findings in their manuscript fully available?

Reviewer #3: Yes

5. Is the manuscript presented in an intelligible fashion and written in standard English?

Reviewer #3: Yes

6. Review Comments to the Author

Reviewer #3: Comments and concerns have been addressed. Repetitive fragment on the text has been removed. The re-evaluation of the meaning of the sequence: "the other strong hypothesis is that the snow covered environment might influence them not only by visual stimulation but also through a reduction in the concentration of volatiles in the air" has been completed

7. PLOS authors have the option to publish the peer review history of their article (what does this mean?). If published, this will include your full peer review and any attached files.

Reviewer #3: No

---

## [Editor Report · Acceptance letter]

21 Dec 2020

PONE-D-20-06816R3 

The Effects of Viewing a Winter Forest Landscape with the Ground and Trees Covered in Snow on the Psychological Relaxation of Young Finnish Adults: A Pilot Study 

Dear Dr. Bielinis:

I'm pleased to inform you that your manuscript has been deemed suitable for publication in PLOS ONE. Congratulations! Your manuscript is now with our production department. 

Kind regards, 

on behalf of

Dr. Marco Innamorati 

Academic Editor

PLOS ONE